# Augmented Reality Based Interactive Cooking Guide

**DOI:** 10.3390/s22218290

**Published:** 2022-10-28

**Authors:** Isaias Majil, Mau-Tsuen Yang, Sophia Yang

**Affiliations:** 1Department of Computer Science & Information Engineering, National Dong Hwa University, Hualien 974301, Taiwan; 2Interdisciplinary Program of Electrical Engineering & Computer Science, National Tsing-Hua University, Hsinchu 300044, Taiwan

**Keywords:** augmented reality, Magic Leap One, smart kitchen, AR cooking

## Abstract

Cooking at home is a critical survival skill. We propose a new cooking assistance system in which a user only needs to wear an all-in-one augmented reality (AR) headset without having to install any external sensors or devices in the kitchen. Utilizing the built-in camera and cutting-edge computer vision (CV) technology, the user can direct the AR headset to recognize available food ingredients by simply looking at them. Based on the types of the recognized food ingredients, suitable recipes are suggested accordingly. A step-by-step video tutorial providing details of the selected recipe is then displayed with the AR glasses. The user can conveniently interact with the proposed system using eight kinds of natural hand gestures without needing to touch any devices throughout the entire cooking process. Compared with the deep learning models ResNet and ResNeXt, experimental results show that the YOLOv5 achieves lower accuracy for ingredient recognition, but it can locate and classify multiple ingredients in one shot and make the scanning process easier for users. Twenty participants test the prototype system and provide feedback via two questionnaires. Based on the analysis results, 19 of the 20 participants would recommend others to use the proposed system, and all participants are overall satisfied with the prototype system.

## 1. Introduction

Home cooking can be both a healthy hobby and a sustainable activity. Nevertheless, the traditional cooking experience tends to be tedious and unenjoyable, especially for those who are unskilled in the kitchen. Using recipes has been the traditional way to teach and learn how to cook, but this may lead to several practical issues. The first issue is to find recipes for ingredients you already have in the house. Locating available ingredients and matching them with the right recipes is not straightforward, but it is an eco-friendly practice of reducing food waste. Another issue is to read and follow a recipe in the process of cooking. Switching back and forth between preparing the food and reading the recipe is neither convenient nor safe. 

Fortunately, augmented reality (AR) technology can superimpose a virtual demo video on an actual kitchen scene so the user can follow a recipe more easily. In addition, computer vision (CV) technology can sense the actual kitchen environment so that food ingredients in a refrigerator or cabinet can be detected and recognized automatically. Best of all, both AR and CV technologies can be integrated in an all-in-one AR headset with a built-in camera to significantly enhance the cooking experience.

Numerous apps are available to help with cooking by suggesting recipes or providing detailed instructions using either a tablet or smartphone in kitchen. However, these modern gadgets can neither perceive the actual environment nor follow the user around the kitchen. Alternatively, some novel smart kitchens are designed to save time and energy spent on cooking. Typically, they demand that users install several Internet-connected sensors, cameras, projectors, and displays in the kitchen, which makes their deployment challenging and expensive. 

To improve the cooking experience, we propose an AR cooking assistance system that simply requires the user to put on an all-in-one AR headset called Magic Leap One [1]. As shown in Figure 1, the user can command the built-in camera on the AR headset to locate and classify food ingredients automatically by merely glancing at them. Accordingly, corresponding recipes are suggested based on the types of the recognized food ingredients. Then a step-by-step video tutorial of the chosen recipe is displayed using the AR glasses, without blocking the real view of the kitchen. The whole process can be controlled by hand gestures, meaning that users do not need to hold any remote controller or touch any physical button. The proposed AR cooking system aims to provide users with an easy-to-use interactive recipe and an easy-to-understand cooking guide through the use of an AR headset.

Cooking at home comes with numerous challenges such as classifying food ingredients, searching for potential recipes, and following the recipes for cooking. The contribution of this paper is finding the solutions for these issues, implementing the algorithms, and integrating them in an all-in-one AR headset. The proposed AR cooking assistance system has the following advantages:Users do not need to install any external devices in the kitchen. All they need is an all-in-one AR headset that costs about 550 USD.Without holding a smartphone or tablet to aim at a specific food ingredient, the user can direct the built-in camera on the AR headset to detect and recognize multiple food ingredients by simply looking at them.No matter where the user moves in the kitchen, the demonstration video is always in the field of view (FOV) of the user.The demonstration video is superimposed on the real-world scene without blocking the line of sight of the actual cooking.Without holding any remote controller or touching any physical button, the user can control the proposed system through non-touch interaction using natural hand gestures.

The remaining parts of the paper are structured in six sections. Section 2 discusses the state-of-the-art works related to smart kitchen or AR cooking. Section 3 describes the methodology and implementation of the proposed AR cooking system. Section 4 explains the deep learning models for food ingredient detection and recognition. Section 5 presents the user study of the prototype system. Section 6 discusses the analysis results of users’ questionnaires. Lastly, the conclusions and future works are reported in Section 7. 

## 2. Related Work

Plenty of research works and projects have been proposed, such as a smart kitchen based on the Internet of Things (IoT) [2], user centric smart kitchen [3], AREasyCooking [4], and CounterIntelligence [5]. Projects regarding smart kitchens typically require the use of a large number of sensors to detect kitchen appliances, ingredients, and other objects that are necessary for cooking [6]. These sensors include temperature sensors, humidity sensors, IR flame sensors, and passive infrared sensors. All these sensors are usually connected to Internet so the smart kitchen can be controlled with a user’s smartphone for easier access. Similarly, other IoT-based smart kitchens have been proposed to ensure safety through the detection of liquefied petroleum gas [7,8] or CO_2_ [9] leaks, as well as fire monitoring [10]. Nevertheless, the requirement of numerous Internet-connected sensors means that the IoT-based smart kitchen has not become very popular.

The goal of a smart kitchen is to take away the stress of cooking [11]. A user-centric smart kitchen [3] is a support cooking system that consists of three modules: tracking food, identifying food materials, and recognizing cooking actions. Three optical cameras are used to identify the food materials while a thermal camera is used to monitor the stove’s heating condition. Besides recognizing the environment, these sensors are also used to recognize cooking actions. Both materials and cooking actions are analyzed to determine the current cooking status. The end of the cooking task is determined by recognizing the final cooking action.

Another research direction for cooking assistance is AR. AREasyCooking [4] is an application that uses AR to help people to cook by utilizing eye and voice controls. The first process is to recognize an ingredient based on its appearance using a neural network model or scan the barcode on a canned food. Then, recipes are selected from a database based on the detected ingredients. The recipes are in a text format and can be supplemented with images or videos. Voice control and eye control are used to interact with the video aids. Some keywords are used to trigger certain actions through voice recognition. 

In addition, Hasada et al. [12] focus on three types of cookware and compare three AR display methods: images with text, video, and 3D animation, using Microsoft HoloLens [13]. Zhai et al. [14] identify five major aspects with which cooking novices need assistance: food preparation, cooking method, ingredient usage, time control, and process understanding. Five corresponding auxiliary guidance tools are displayed using the HoloLens to assist unskilled users in cooking. Alternatively, Reisinho et al. [15] present a serious hybrid board game to enhance children’s cooking skills by simulating the cooking processes through AR. Ricci et al. [16] design an AR-enabled kitchen machine to guide users in the cooking activity using the HoloLens 2. Lastly, Styliaras [17] reviews the use of AR in food analysis and promotion through products and orders. Similarly, Chai et al. [18] review food-related applications and research works using AR/MR in the food industry.

Smart kitchens and AR cooking are two different ways to make cooking easier and more effective, but these ideas can also be integrated to build a more complete system. CounterIntelligence [5] is an AR smart kitchen combining features of an AR cooking environment with those of a smart kitchen. AR features are applied via the use of projectors, while the smart kitchen features are implemented through the use of LEDs and infrared thermometers. Contents inside a refrigerator are projected outside, and an interactive step-by-step recipe is projected onto kitchen cabinets. LEDs are deployed in order to find cooking equipment more easily, and the infrared thermometers are used to display the temperature of running water in a sink. Alternatively, Balaji et al. [19] propose a smart kitchen wardrobe that can monitor and detect grocery products inside. Samsung focuses on the design of smart refrigerators, called food AI [20], combining AI and image recognition. The smart refrigerator keeps track of the items inside and their expiration dates, thus helping users to solve the problem of waste food. 

Table 1 compares the pros and cons of the proposed system and nine other related works. Compared to other smart kitchens or AR cooking methods, the proposed system requires only a pair of all-in-one AR glasses, called Magic Leap One [1], without the need to install any external sensors or devices in the kitchen. In addition, the user can direct the built-in camera on the AR headset to locate and classify food ingredients by just looking at them. Then, suitable recipes are suggested based on the types of the recognized food ingredients. Subsequently, the AR glasses display a step-by-step video that demonstrates each cooking step in the chosen recipe. No matter where the user moves in the kitchen, the demonstration video is always in the field of view of the user without blocking the real kitchen scene. Best of all, the whole process can be controlled by natural hand gestures so that users can cook without needing to hold any device or controller in their hands. By using the proposed non-touch interactive system, users can make sure both hands are clean during the whole process of cooking.

## 3. Implementation Methods

The Magic Leap One is the target AR headset for the proposed cooking assistance system. A PC with Windows 10 is used as the development platform of the proposed AR cooking application. The software engine used to create the proposed application is the Unity 2020.1.6f1 because of its cross-platform compatibility with the Magic Leap One. The Lumin SDK [1] is adopted to connect the Unity and the Magic Leap One to create an AR interface based on hand gesture recognition. From the user’s perspective, the proposed cooking assistance system requires only a pair of all-in-one AR glasses without the need to install any external sensors or devices in the kitchen. The total cost of the solution is about the price of the Magic Leap One, which has been reduced to 550 USD in 2022.

As shown in Figure 2, the methodology of the proposed AR cooking system can be fundamentally divided into three main phases: food ingredient scanning, recipe recommendation, and a step-by-step cooking video tutorial. In the first phase, a user can simply glance over food ingredients on the kitchen table or in the refrigerator, and the built-in camera on the AR headset can detect and recognize them automatically. In the second phase, a list of best-fit recipes is provided and sorted according to the proportion of essential food ingredients that are available. Then, the user can choose a recipe from the list. In the third phase, the AR glasses are utilized to display a step-by-step recipe with a video tutorial on how to perform each cooking step. To guarantee that the user’s hands are clean throughout the cooking process, all three phases of the proposed AR cooking system can be controlled via the user’s natural hand gestures in real-time, without the need to hold a controller in their hand. 

Figure 3 shows the complete flowchart of the proposed AR cooking system. At the beginning, users can choose between two options on the title screen via hand gestures. The first option is for users that already have a recipe in mind. In this case, a list of all available recipes is provided, and the user can directly choose a recipe from the complete recipe list. Another option is for users who want to cook using food ingredients available in the house. In this case, the user needs to scan available food ingredients on the kitchen table or in the fridge using CV technology. The front view of the user is captured by the built-in camera on the AR headset and analyzed by a deep learning approach to locate and classify food ingredients automatically. The training and recognition of the deep learning models are thoroughly explained in Section 4. The scan process can be repeated until sufficient food ingredients are recognized. In the next phase, the proposed system suggests a list of recipes according to the types of the recognized food ingredients. 

Once sufficient food ingredients are detected and recognized, the user is provided with a list of suggested recipes based on the recognized ingredients. The list consists of all recipes with at least one required main ingredient detected and is sorted according to the proportion that is computed as the number of the available essential ingredients divided by the number of the required ingredients: $$\mathrm{Proportion}\;=\;\frac{\left(\mathrm{Total}\;\mathrm{main}\;\mathrm{ingredients}\;\mathrm{recognized}\right)\;\cap \;\left(\mathrm{Total}\;\mathrm{main}\;\mathrm{ingredients}\;\mathrm{required}\right)}{\mathrm{Total}\;\mathrm{main}\;\mathrm{ingredients}\;\mathrm{required}}\;*100\%$$

Figure 4 provides an example in the case of only eggs being detected. Minor ingredients, such as flour, oil, and seasoning, are assumed to be always available. The proportion of each recipe is computed and shown on the right side of the recipe name. Using the cake recipe as an example, eggs are the only main ingredients needed, hence representing a proportion of 100%. On the other hand, the main ingredients for the omelette recipe are eggs, green onions, and spam—a proportion of 33%. Then, the user can choose a recipe from the list by hand gestures. The hand gesture is different for each recipe, so an icon of the corresponding gesture is displayed on the left side of the recipe name. 

After selecting a recipe, the user is offered an overall recipe screen with a picture of the finished product and the detailed instructions, as shown in Figure 5. With the whole picture in mind, the user can then start practicing the recipe by following the step-by-step procedures. In each cooking step, a video tutorial is provided to help the user prepare meals. As shown in Figure 6, a series of steps is displayed on top of the AR headset’s field of view with a red highlight on the current working step. A corresponding video clip demonstrates how to carry out the cooking tasks in each step. The video window’s default location is in the upper middle of the AR headset’s field of view, which always follows the user’s head movements. The AR headset automatically blends virtual foreground and real background images together so the video window is semi-transparent on the foreground, and the user can see a little bit of the real scene beneath. Optionally, the user can choose if they want to move the video window to any other designated position to prevent the video window from blocking the real view of the kitchen scene behind it. At all times, the user can decide when to move on to the next step of the recipe via hand gestures.

Hands are usually busy and must remain clean in the process of cooking. Instead of using a touch screen or holding a controller in the hand, bare-hand gestures are recognized to control the cooking tutorial and the video playback in the proposed system. An API provided by the Magic Leap One, called Lumin SDK [1], is utilized to classify hand gestures on images captured by the built-in camera on the AR headset. It supports eight discrete hand gestures from either hand, including “*C-Gesture*”, “*L-Gesture*”, *“Open Hand-Gesture”, “Finger Up-Gesture”, “Fist-Gesture”, “OK-Gesture”, “Pinch-Gesture”,* and *“Thumbs Up-Gesture”*. In addition, it also includes a state where no hand gesture is recognized. As shown in Table 2, the “*Open Hand-Gesture*” is used to lock the recipe window on any designated corner to prevent it from blocking the view of the real environment. The “*OK-Gesture*” is used to trigger the scanning of food ingredients. It is also used in case the user wants to move on to the next step of the recipe. In contrast, the “*L-Gesture*” is used if the user wants to move back to the previous step of the recipe. The “*Pinch-Gesture*” can be used to click on buttons or to select a recipe from the recipe list. In addition, it can be used to move the step-by-step recipe until it is locked into the right place. The “*Fist-Gesture*” stops a video from playing, and the “*Finger Up-Gesture*” plays the corresponding video along with the recipe. The “*Thumbs Up-Gesture*” can be used to take a picture while in the scanning screen for food ingredient recognition and can be used in the title screen to select the button to open the recipe list. It is also used in the recipe list menu to start a step-by-step recipe. Finally, the “*C-Gesture*” is reserved to exit the system after the cooking is finished. By using these hand gestures, the proposed AR cooking guide is a fully non-touch interactive system.

## 4. Deep Learning Model for Food Ingredient Recognition

With the advance of CV technology, many deep learning models based on the CNN (Convolutional Neural Network) can be utilized to recognize food ingredients in an image. Usually, the models assume that the target object is the only subject located at the center of the image. To detect and recognize numerous objects with multiple categories in an image, it is necessary to apply models to the image at multiple locations and scales. A location and scale with a high prediction score are considered a detection. This repetitive process makes them inefficient and inconvenient for food ingredient scanning in our application.

On the other hand, the deep learning model called YOLO (You Only Look Once) [21] is an object detection algorithm that applies a single CNN to the entire image. It divides the image into regions and predicts bounding boxes and probabilities for each region. The YOLO model returns not only prediction scores for each category but also a few bounding boxes and their confidence scores. The merits of the YOLO model are the real-time speed and the capability to locate numerous objects and classify multiple categories at the same time. For this reason, the proposed AR cooking system adopts the latest version of the YOLO, called YOLOv5 [22]. 

The YOLO models have been incrementally improved over earlier versions; thus, the network architecture of YOLOv5 is highly complicated. As shown in Figure 7, it can be generally divided into three stages: the backbone, the neck, and the head. First, the backbone of the YOLOv5 incorporates the cross-stage partial network (CSPNet) [23] into the Darknet for feature extraction. The focus layer is designed to reduce layers, parameters, and memory, as well as to increase the speed of the forward and backward propagation. The spatial pyramid pooling layer is used to remove the fixed size constraint of the network. Second, the neck of YOLOv5 adopts the path aggregation network (PANet) [24] to boost information flow for feature fusion. It can increase the location accuracy of the detected object by utilizing accurate localization signals in lower layers. Third, the head of YOLOv5 generates three different sizes of feature maps to predict classes and bounding boxes in multiple scales.

In the training stage, we rely on a food ingredient dataset, called Q-100 [25], consisting of 905 images which are divided into 3 parts: training, validation, and testing. The training part comprises 631 images (70%), the validation part comprises 179 images (20%), and the testing part comprises 95 images (10%). The dataset comes with an average of 3.8 annotations per image, with a total of 3408 annotations. As shown in Figure 8, there are 11 classes in this dataset including sprout, beef, chicken, egg, pork, garlic, onion, kimchi, onion, potato, and spam. The training process is performed using Python on Jupyter.

In the recognition stage, the constructed network with pre-trained weights can be used directly for food ingredient detection and recognition. The DNN module in the OpenCV supports YOLOv5. However, Unity only supports scripts written in C# and cannot natively run the OpenCV code in C and C++. A third-party asset, called *OpenCV for Unity* [26], is employed to integrate OpenCV with Unity so the recognition of food ingredients can be carried out based on the pre-trained model. 

The YOLOv5 model is trained on the Q-100 food ingredient dataset for 100 epochs, and it takes 9.5 h to complete. The training time can be shortened significantly if a powerful GPU is used instead of only a CPU. Figure 9 demonstrates the results of the training and validation of the YOLOv5 on the Q-100 food ingredient dataset. The upper row shows the results of training, while the lower row shows the results of validation. The horizontal axis of each subgraph represents the number of epochs. The vertical axis of each subgraph represents the *box_loss* (error of location), *obj_loss* (error of detection), *cls_loss* (error of classification), *precision*, *recall,* and *mAP* (mean average precision), respectively.
*precision* = *True Positives*/(*True Positives* + *False Positives*)
*recall* = *True Positives*/(*True Positives* + *False Negatives*)
$$mAP=\frac{1}{n}\sum_{k=1}^{n}\;AP_{k}\;,\;where\;AP_{k}\;=\;\text{average precision of class k}$$
*F-score* = *2***precision***recall*/(*precision*+*recall*)

A *true positive* is a correct detection made by the model, a *false positive* is a detection made by the model that turned out to be incorrect, and a *false negative* is when something is not detected or is missed. A model is good if it has high *precision* and high *recall*. A trade-off between *precision* and *recall* is determined heuristically in the proposed application.

Figure 10 shows the confusion matrix of the recognition over 11 types of food ingredients. We can see that eggs can be detected with the highest accuracy of 96%. Most other food ingredients can be recognized with an accuracy well above 60%, except for chicken, pork, and beef. These meat ingredients usually come in different shapes and a variety of packages, thus resulting in lower accuracy. There is a trade-off between *precision* and *recall*. To more precisely evaluate accuracy, an *F-score* is computed as the harmonic mean of *precision* and *recall*. Overall, the YOLOv5 achieves an *F-score* of 0.61. To improve the accuracy of the recognition, we have tried other deep learning models such as ResNet [27] and ResNeXt [28]. Table 3 compares the performance, speed, delay, and capability of these deep learning models. Generally, ResNet and ResNeXt models improve the accuracy with an *F-score* of 0.78. However, they can only classify one ingredient at a time, and the ingredient is expected to be the only subject in the image. It is troublesome and time-consuming for users to aim at each food ingredient and classify them one after another. On the other hand, YOLOv5 can detect and recognize multiple food ingredients at the same time. To make the food scanning process more user-friendly, our AR cooking system adopts YOLOv5 to locate and classify food ingredients efficiently.

For simplicity, the prototype AR cooking system currently focuses on vegetarian recipes. Figure 11 shows some results of the detection and recognition of food ingredients using YOLOv5. It can be seen that YOLOv5 can locate and classify multiple ingredients most of the time. However, there are still times when some ingredients are not detected, such as the partially occluded onions, and some ingredients are classified wrongly, such as the confusion between a potato and an egg.

## 5. Case Study

Twenty people participated in the testing of the prototype system and gave feedback regarding how easy the system was to use via a usability questionnaire (UQ as shown in Appendix B). Of these 20 participants, 12 were male and 8 were female. Their technical skills and backgrounds were recorded via another background questionnaire (BQ as shown in Appendix A). Before real cooking, participants were given a preparation time of 10 min to become familiar with the Magic Leap One headset, the real kitchen, and the cooking equipment. They were given a printout (Table 2) of eight hand gestures that can be recognized, as well as their functions, so they did not need to memorize all the hand gestures. Afterwards, the participants were asked to wear the AR headset with the proposed AR cooking system installed and proceeded to use it for cooking assistance to prepare meals. To ensure fairness, everyone was asked to follow the same recipe for white cake. Participants were given ingredients to cook, and as an incentive, the finished products (cakes) were theirs to keep. Due to the limited number of AR devices, one participant at a time used the proposed AR cooking system, and it took about an hour for the cooking task to be completed.

Before participants used the proposed system (usually, while they waited for their turn), they were asked to fill out a background questionnaire (BQ as shown in Appendix A). This questionnaire was used to gauge how proficient they were in cooking and their experience with AR. After they completed the cooking task using the proposed system, they were requested to fill out a usability questionnaire (UQ as shown in Appendix B). This questionnaire was used to measure the ease of use of the proposed system. All questions in both questionnaires were designed according to the five-point Likert scale, which contains five response options (strongly disagree, disagree, neutral, agree, strongly agree). In total, each participant filled out two questionnaires with optional open feedbacks and suggestions on how the system can be improved.

After the results from both background questionnaires (BQ) and usability questionnaires (UQ) were collected, we made statistical charts in order to get a more concrete idea of the participant’s answers. By assigning five rating scores (1~5) to the five response options (strongly disagree, disagree, neutral, agree, strongly agree) in the five-point Likert scale, Figure 12 shows the mean and confidence interval (alpha = 0.05) for each question in the background questionnaire. Half of the participants either agreed or strongly agreed to having an extensive knowledge of cooking (BQ1), and more than half of the participants cooked often (BQ2). The majority of the participants were confident in following a simple recipe, while only one participant disagreed with this (BQ4). We can also see that more than half of the participants enjoyed homemade meals more than take-out food (BQ5). However, half of participants bought takeout more than they made homemade food (BQ9). 

A correlation analysis was conducted over the questions in the background questionnaire. A correlation coefficient (a value between −1 and 1) represents how strongly two variables are related to each other. As a correlation coefficient approaches 1, it indicates that there is a positive correlation. This implies that as one variable increases, so does the other. The opposite holds true as well—as a correlation coefficient approaches −1, it indicates that there is a negative correlation, which implies that as one variable increases, the other decreases. The most significant positively correlations (0.98) were for BQ5, “*I prefer eating homemade food over eating takeout*”, and BQ6, “*I enjoy cooking*”. This suggests that when one enjoys cooking more, one prefers to eat more homemade food than take-out food.

After assigning five rating scores (1~5) to the five response options (strongly disagree, disagree, neutral, agree, strongly agree) in the five-point Likert scale, Figure 13 shows the mean and confidence interval (alpha = 0.05) for each question in the usability questionnaire. Most (19 of the 20) participants agreed or strongly agreed that the proposed system was easy to use (UQ1), while 18 of the participants agreed or strongly agreed that it was easy to learn how to use the system (UQ3). In addition, the majority of participants agreed or strongly agreed that they would use the system again (UQ2). Most of the participants did not feel any discomfort or awkwardness when using the system (UQ9). All 20 participants were satisfied with the end product of the white cake (UQ12), and they were also satisfied with the proposed AR cooking system (UQ13). Meanwhile, 19 participants would definitely recommend the system to others (UQ11). 

A correlation analysis was conducted with the questions in the usability questionnaire. This indicated that UQ11, “*I would recommend the system to others*”, and UQ12, “*I am satisfied with the end product*”, had a perfect positive correlation coefficient. This suggests that if a user was satisfied with what they had cooked, they were more willing to recommend the system to others. In addition, UQ12, “*I am satisfied with the end product*”, and UQ13, “*Overall, I am satisfied with the system*”, had a perfect positive correlation. This implies that if a user was satisfied with what they had cooked, they were satisfied with the system as well.

Finally, we also analyzed the correlation between participants’ cooking background and their experience with using the proposed AR cooking system. The most significant correlation coefficient (0.97) was for BQ4 “I am confident in following a simple recipe” and UQ2 “I would use the system again”. This suggests that the more confident the user was in following a recipe, the higher the chance they would like to use the proposed system again, mainly because the proposed system is a step-by-step recipe guide. However, if they did not want to use the proposed system again, that means they might have developed a negative view of the cooking guide system, and hence their confidence in following a recipe may be reduced. On the other hand, the most significant negative correlation coefficient (-0.94) was for BQ6, “I enjoy cooking”, and UQ5, “I needed prior knowledge in order to use the system”. If no prior knowledge is required to use the system, this means the system is easy to use, and if the system is easy to use, the user will enjoy cooking more. This matches the goal of the proposed system to make people enjoy cooking. The opposite is also true: if one needs prior knowledge in order to use the system, this means the system is hard to use, and thus the user will not enjoy cooking. 

## 6. Discussion

Instead of dining out or buying ready-to-eat food, cooking your own meal is cheaper and healthier. Home-cooked meals gives you greater control over the ingredients and calories in your meals, thus improving weight management, fulfilling personal needs, and reducing illness risk. According to the feedbacks from the received questionnaires, we confirm that the proposed AR cooking guide system is feasible and practical for cooking assistance. Most participants had no trouble learning and using the proposed system. In total, 19 of the 20 participants would recommend the system to others to use (UQ11). All participants were satisfied with their end products from their baking (UQ12), and all participants were overall satisfied with the system (UQ13) (either strongly agreeing or agreeing). 

Regarding the optional feedbacks, most participants stated that once they were used to the hand gestures, the system gradually became easier to use as time goes on. In addition, the demonstration video for each cooking step was helpful because worded steps can be a bit vague. Several participants believe that making the whole process non-touch is the best feature because having clean hands is an important part of cooking. A non-touch interactive system assures users that their hands touch only the food ingredients, and they can cook while not having to touch anything else. Some participants express that being able to lock the video window in a designated position is another handy feature. This way, it does not interfere with the field of view of the real scene behind, and the user can look back and forth between the virtual video tutorial and their physical working area in order to cook efficiently. In particular, two participants reported that they prefered to be able to minimize the video window in certain cooking steps. 

Regarding the optional suggestions for improvement, some participants felt that the hand gesture recognition was too sensitive. Sometimes, the system recognized hand gestures accidentally when the participant was actually doing something else, which resulted in unnecessary hassles. In addition, since some hand gestures look alike, the recognition system occasionally misidentified a hand gesture as something else and executed the wrong function. The gesture recognition needs to be more intuitive and less sensitive. A careful tuning of the thresholds could be helpful to achieve a better trade-off between precision and recall. Moreover, instead of recognizing static hand gestures solely based on an image, recognizing dynamic hand actions based on a short-term video has the potential to reduce confusion and should be more robust and reliable. Furthermore, two participants suggested having the system recognize both hands instead of one hand, which can lead to more combinations of gestures that are essential for cooking action recognition. Besides, a participant also suggested some recipe steps could be improved to sound less vague, especially in terms of measurements. A participant also mentioned that “softer colors” would be a better choice to improve the visualization of the interface. 

In addition to the 10 min preparation time, participants took about 50 min to follow all the steps, mix the ingredients, and bake the cake in the oven, all assisted by the proposed AR cooking system. It is interesting to note that one hour is normally the time it takes for an experienced baker to bake a cake. Even if less than half of the participants had experience of baking a cake (BQ8), the proposed AR cooking system was useful and effective in helping unskilled people to complete the cooking task within the expected time limit. All participants were successful in the baking of their cakes. No destructive mistakes occurred during our experiments. Even if a few participants needed to restart the demo video in some cooking steps due to misunderstandings of the procedures, all participants were satisfied with the cake they made.

One problem encountered in our experiment is that users could not wear prescription glasses with the original Magic Leap One. According to the website of Magic Leap, a prescription insert is available, but it is custom-made for each user and requires additional purchase. Another problem is the overheating of the AR headset with prolonged use, which can be felt by the user wearing the headset and possibly causes dizziness for some people. 

## 7. Conclusions

We propose a new prototype AR system for cooking assistance in which a user only needs a pair of all-in-one AR glasses without having to install any external devices or sensors in the kitchen. We try to overcome some common troubles in cooking, implement the algorithms, and integrate them in an all-in-one AR headset. The user can direct the AR headset’s built-in camera to detect and recognize food ingredients by simply glancing over them in the refrigerator or on the kitchen table. Accordingly, the types of the recognized food ingredients are used to match appropriate recipes. Then, the proposed system provides and displays interactive demo videos on how to carry out each cooking step in the chosen recipe. All processes can be controlled via the user’s natural hand gestures in real-time, without the need to hold a controller in the hand. Compared with the deep learning models ResNet and ResNeXt, YOLOv5 achieves lower accuracy for ingredient recognition, but it can locate and classify multiple ingredients at the same time and thus greatly simplify the scanning process for users. Twenty people participated in the testing of the prototype system, provided feedback via questionnaires, and suggested improvements. All participants were overall satisfied with the prototype system, and 19 of the 20 participants would recommend others to use it; hence, the usability of the proposed AR cooking assistance system is confirmed.

The prototype could be extended in the future by including more interactive recipes. The list of suggested recipes could also provide more information such as nutrition facts and calorie counts. In addition, implementing a scalable database to manage the addition of recipes for better tracking and storing should make the system more complete. Moreover, the more accurate recognition of a wider variety of food ingredients is another potential area for future research. Finally, the system could be enhanced by recognizing dynamic hand gestures, monitoring cooking actions, detecting procedural mistakes, and guiding users to prevent or recover from potential failure.

## Figures and Tables

**Figure 1 sensors-22-08290-f001:**
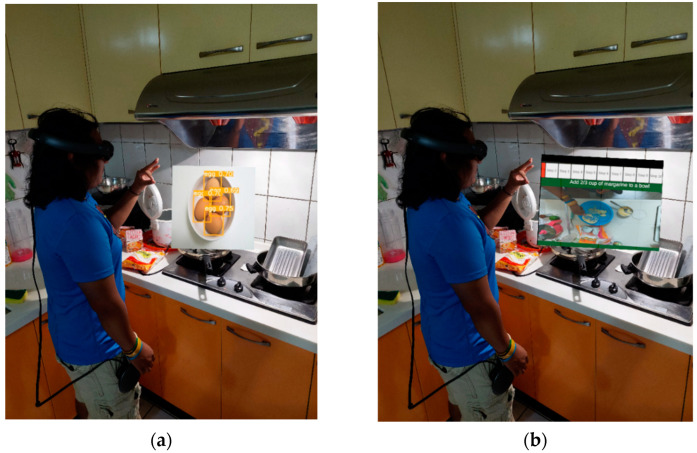
Proposed AR cooking assistance system. (**a**) Ingredient recognition by a built-in camera on AR headset, (**b**) interactive step-by-step demo video controlled by natural hand gestures.

**Figure 2 sensors-22-08290-f002:**
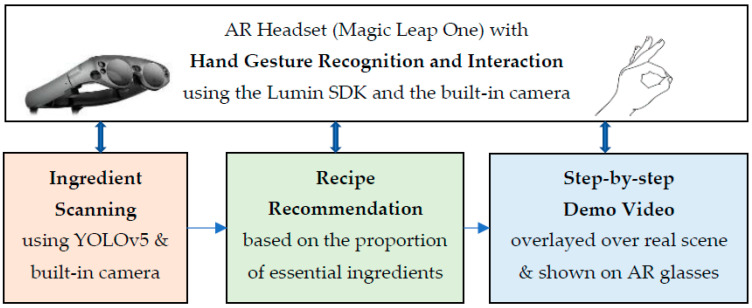
Three main phases of the proposed AR cooking system: food ingredient scanning, recipe recommendation, and step-by-step cooking video tutorial. The whole process can be controlled via hand gestures.

**Figure 3 sensors-22-08290-f003:**
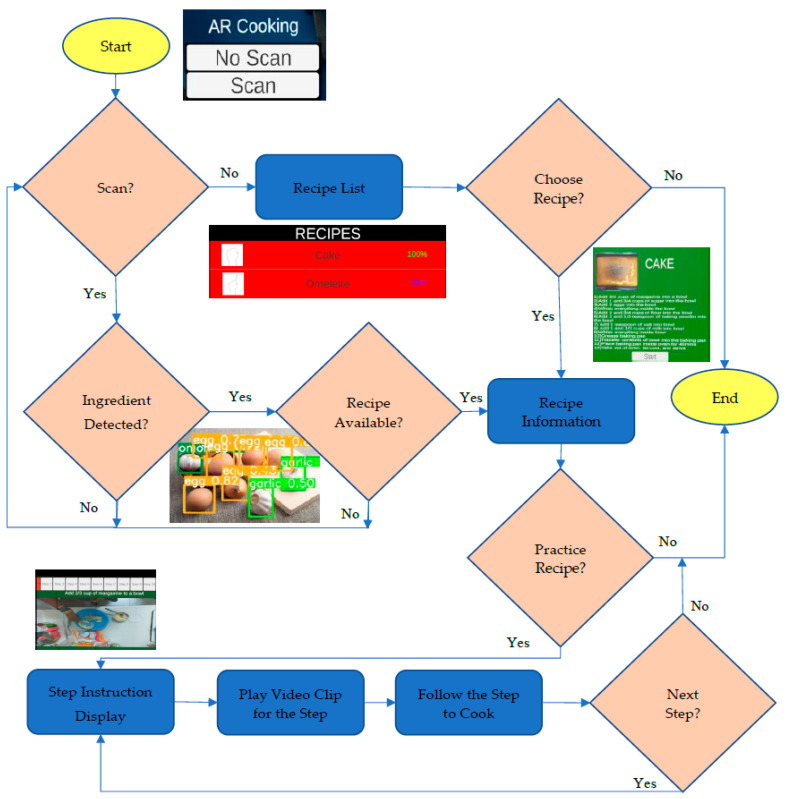
Flowchart of the proposed AR cooking assistance system.

**Figure 4 sensors-22-08290-f004:**
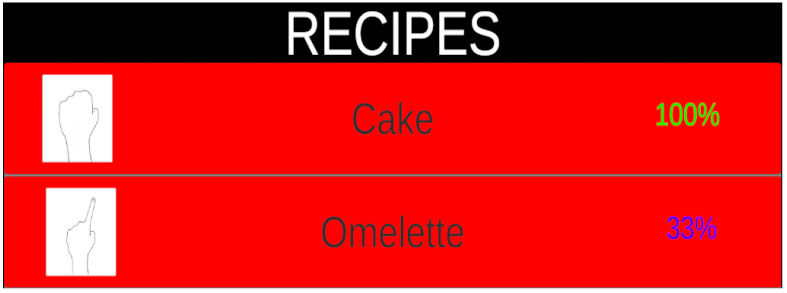
List of recommended recipes sorted according to the proportion of essential ingredients that are available.

**Figure 5 sensors-22-08290-f005:**
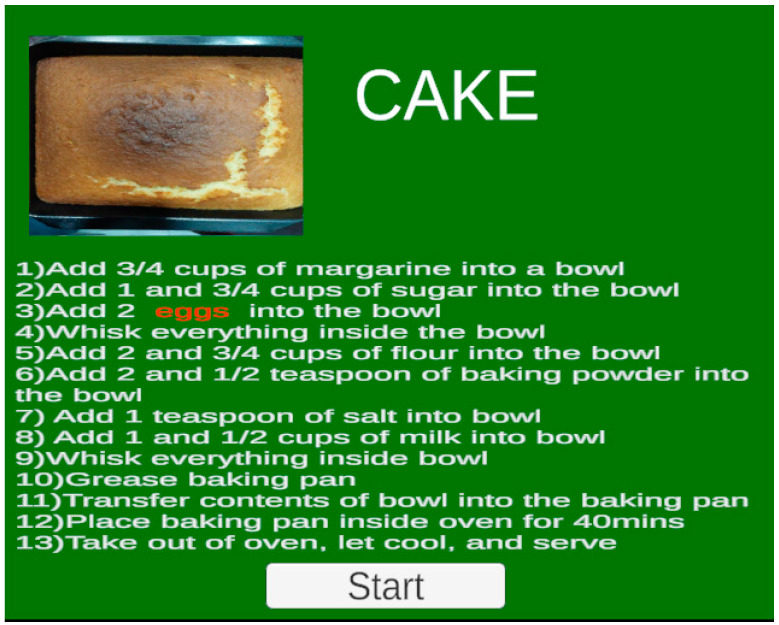
Overall screen showing the recipe for white cake.

**Figure 6 sensors-22-08290-f006:**
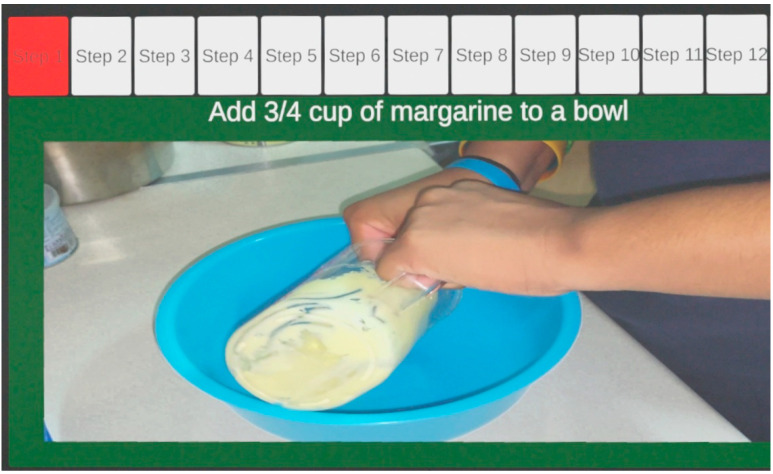
Step-by-step video tutorial for the recipe for white cake. The video clip is semi-transparent on the foreground so the user can see a little bit of the real scene (tiles on kitchen wall) beneath.

**Figure 7 sensors-22-08290-f007:**
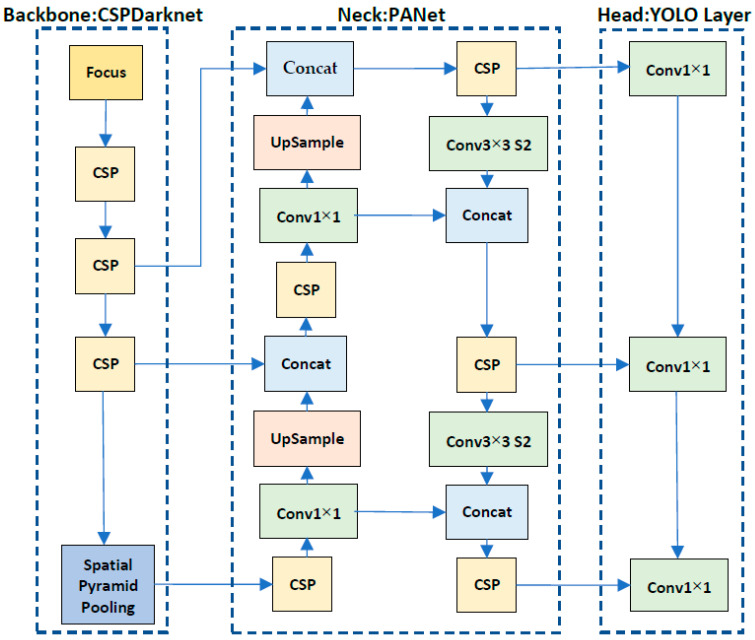
Network architecture of YOLOv5 with three stages: the backbone for feature extraction, the neck for feature fusion, and the head for object prediction.

**Figure 8 sensors-22-08290-f008:**
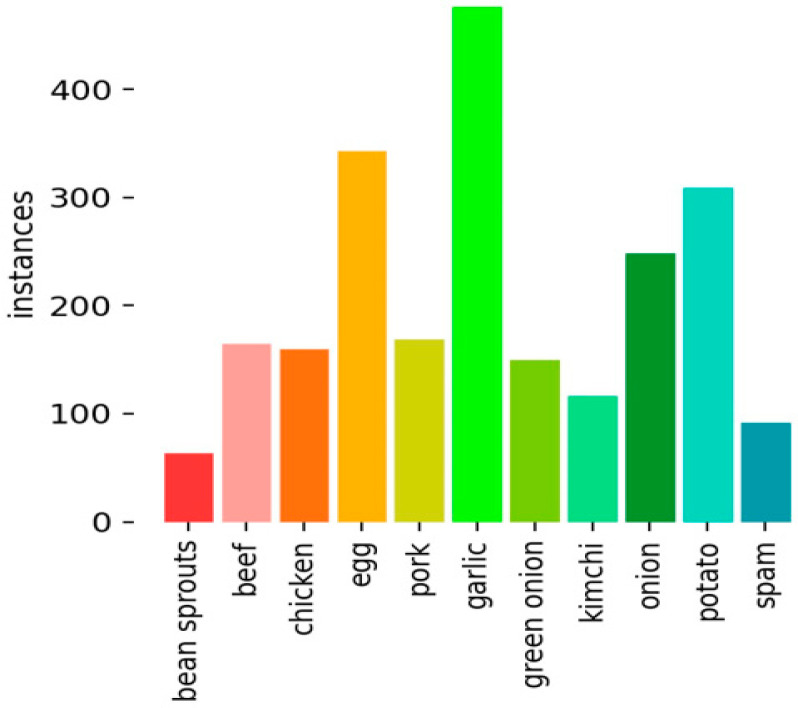
Training dataset containing 11 food ingredients.

**Figure 9 sensors-22-08290-f009:**
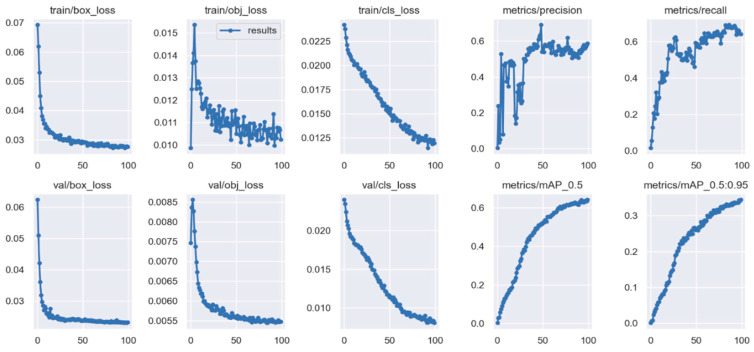
YOLOv5 results of training (upper row) and validation (lower row).

**Figure 10 sensors-22-08290-f010:**
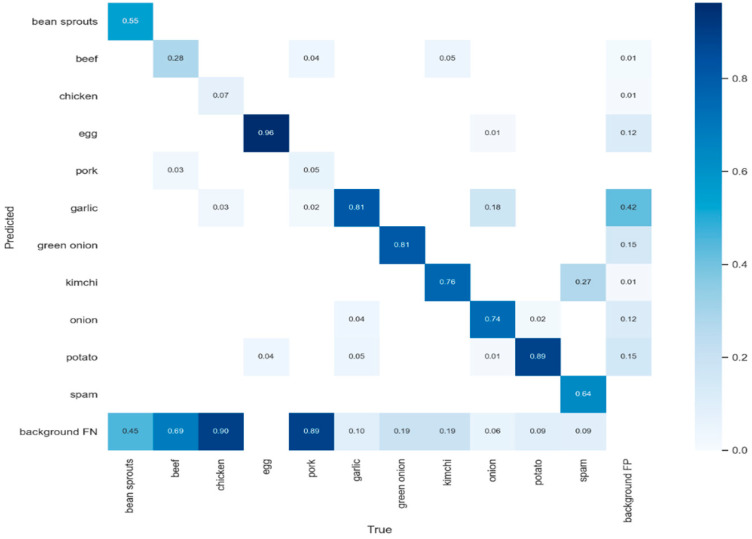
Recognition accuracy and confusion matrix of 11 food ingredients.

**Figure 11 sensors-22-08290-f011:**
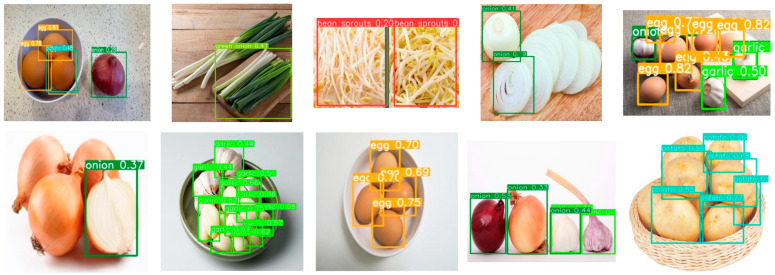
Results of YOLOv5 detection and classification of food ingredients.

**Figure 12 sensors-22-08290-f012:**
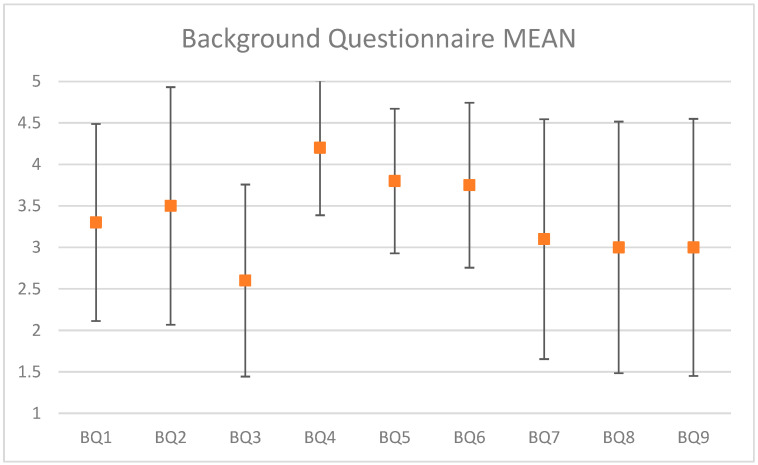
Background questionnaire results: mean and confidence interval (alpha = 0.05).

**Figure 13 sensors-22-08290-f013:**
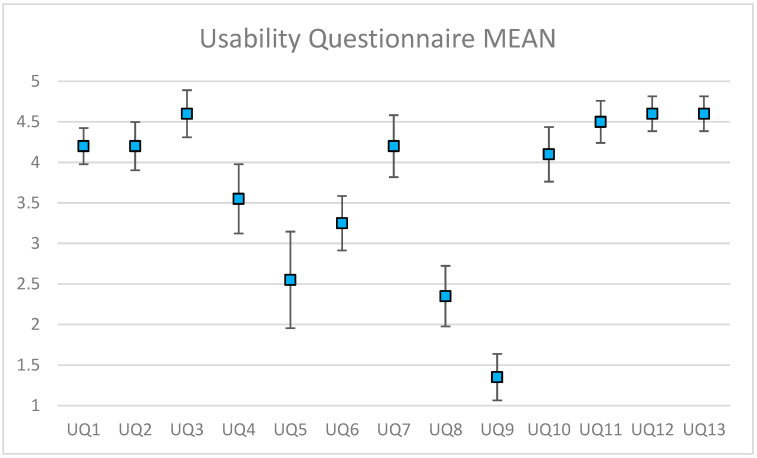
Usability questionnaire results: mean and confidence interval (alpha = 0.05).

**Table 1 sensors-22-08290-t001:** Pros and cons of the proposed research and nine other related works.

Project	Type	Hardware	Pros	Cons
User-Centric Smart Kitchen [3]	Smart Kitchen	Three optical cameras; one thermal camera	Accuracy recognition	Working area is small; items have to stay in the camera’s FOV
Smart Kitchen using IoT [7]	Smart Kitchen	Lots of sensors for gas, flame, weight, humidity, temperatore; IoT	Gas leakage detection	Need to install many Internet-connected sensors in kitchen
Real-Time Kitchen Monitoring [8]	Smart Kitchen	Many sensors for gas, humidity, temperature; smartphone; Arduino; IoT	Control switches, fans, and lights over Internet	Need to install many Internet-connected sensors in kitchen
IoT based Kitchen [10]	Smart Kitchen	Lots of sensors for gas, temperatore, PIR; Smartphone; IoT	Fire detection; person detection	Need to install many Internet-connected sensors in kitchen
Smart Kitchen Wardrobe [19]	Smart Kitchen	Smartphone; Arduino; IoT	Monitoring the groceries in the cupboard	Need a sensor for each container
Counter Intelligence [5]	AR and Smart Kitchen	Camera, projector; infrared thermometer; LED on handles and faucets	Information projected on physical surface; LED embedded items	LED items can be easy to miss if not in direct line of sight
AREasyCooking [4]	AR	Smartphone; tablet	Voice control; eye control; barcode reader	Lighting affects eye controls; Noise affects voice control
Interactive MR Cooking Assistant [14]	AR	HoloLens	Timeline; timer; demo video; seasoning tips; tick marks	Lack of ingredient recognition and corresponding recipe suggestion
AR Kitchen Machine [16]	AR	HoloLens 2; Tablet	Humanoid avatar with animations	AR markers required for tracking
Proposed Research	AR	Magic Leap One (an all-in-one AR headset, no other device required)	Ingredient recognition; recipe recommendation; step-by-step guide video; hand gesture interaction	Headset overheating; users cannot wear prescription glasses

**Table 2 sensors-22-08290-t002:** Eight hand gestures and their functions.

Gesture	Function
*C-Gesture*	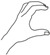	Close the application
*L-Gesture*	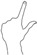	Move to previous step of recipe
*Open Hand-Gesture*	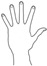	Lock step-by-step recipe into place
*Finger Up-Gesture*	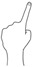	Play video
*Fist-Gesture*	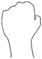	Stop video from playing
*OK-Gesture*	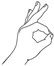	Start scan Move to next step of recipe
*Pinch-Gesture*	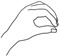	Move recipe when locked to place
*Thumbs Up-Gesture*	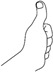	Take picture for scanningOpen recipe list on title screen Start recipe in recipe list menu

**Table 3 sensors-22-08290-t003:** Performance, speed, and capability of three deep learning models.

*Method*	*Precision*	*Recall*	*F-Score*	*Delay (ms)*	*Capability*
ResNet [27]	0.76	0.81	0.78	32	Can only classify one ingredient at a time
ResNeXt [28]	0.75	0.81	0.78	104	
YOLOv5 [22]	0.59	0.64	0.61	125	Can locate and classify multiple ingredients at the same time

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
