# Peer review of "Augmented Reality Based Interactive Cooking Guide"

_sensors, 2022, doi:10.3390/s22218290_

Round 1
Reviewer 1 Report
This paper proposes a system based on augmented reality to navigate and naturally interact with video-based recipes. The application scans a kitchen table, cabinet, or fridge to recognize food ingredients and facilitate the cooking.
According to the submission, the paper needs to be improved, taking into account the following comments:
- There are many typos in the entire manuscript, such as missing spaces, syntaxis and grammar errors, and a combination of different letters (upper and lower cases), prepositions of verbs, among others.
- It is necessary to improve English. There are many mistakes concerning grammatical expressions and verbs. It seems like a “raw translation”. The manuscript is not well written and is difficult its lecture and understand the contribution.
- The abstract must be rewritten and authors must emphasize the contribution, highlining the primary outcomes and reveals.
- Section 1. "Introduction" does not start with a message. It has to move to another section and rewrite the introduction. This section should motivate the readers and center the problem statement.
- It is necessary to modify the actual Section 2 by “Related Work”. This new section must contain the most relevant publications in the state-of-the-art. At the same time, comparison and analysis regarding the literature review with the proposed research work must be presented.
- It is necessary to describe in more detail the three main phases presented in Figure 2. A general diagram of the methodology, which includes the technologies, tools, and algorithms implemented should be incorporated into the manuscript.
- Section 2 “Implementation Methods” is described as a common and familiar narrative. In this case, the section must be described with more formality and theoretical foundations to guarantee the paper's quality. Moreover, the section seems like a user manual. It is more interesting to explain how you implement the gestures shown in Table 2. Present the algorithms and details of the design and implementation.
- Authors must present with more detail and a well-structured deep learning approach to recognize the food ingredients and describe the model. This topic must be explained in an independent subsection.
- It is necessary to present the results in tables, where the precision, recall, and f-measure can be shown with different tests. In addition, a comparison with other approaches state-of-the-art is required to assess the method.
- More results according to the tests are required to present the image recognition (Figure 11).
- I suggest changing the name of Section 4 to “Case Study” instead of “User Study”.
- It is necessary to rewrite Section 6. The main contribution should be highlighted the outcomes well explained.
Reviewer 2 Report
Strengths:
(+) The paper is well-written.
(+) The references are appropriate.
(+) The experiments are convincing.
Weaknesses:
(-) The introduction must be improved.
(-) The related work should be discussed in a separate section.
(-) Experimental evaluation must discuss performance metrics in terms of delay (if any).
Some figure(s) are blur. Authors should either use a higher resolution figure(s) or redo them as vector graphics.
The introduction should clearly explain the key limitations of prior work that are relevant to this paper.
The authors should explain clearly what the differences are between the prior work and the solution presented in this paper.
Some additional experimental setup details are required
Hardware Requirements
Software Requirements
Total cost of solution
Round 2
Reviewer 1 Report
- It is necessary to rewrite again Section 1 "Introduction". It must motivate the audience to read the paper. Moreover, at the end of the section, the authors should provide a brief description of the rest of the sections included in the manuscript.
- References must be updated to research works of 2021-2022 to justify the research field. I encourage authors to make a deep investigation of the state-of-the-art.
- It is necessary to present more detail concerning the learning model for food ingredient recognition. A general schema and foundations of this model are required in Section 4.
